# Physiological Alterations in Relation to Space Flight: The Role of Nutrition

**DOI:** 10.3390/nu14224896

**Published:** 2022-11-19

**Authors:** Stavroula Chaloulakou, Kalliopi Anna Poulia, Dimitrios Karayiannis

**Affiliations:** 1Department of Clinical Nutrition, “Evangelismos” General Hospital of Athens, 10676 Athens, Greece; 2Laboratory of Dietetics and Quality of Life, Department of Food Science & Human Nutrition, School of Food and Nutritional Sciences, Agricultural University of Athens, 11855 Athens, Greece

**Keywords:** space flight, muscle loss, muscle mass, sarcopenia, osteoporosis

## Abstract

Astronauts exhibit several pathophysiological changes due to a variety of stressors related to the space environment, including microgravity, space radiation, isolation, and confinement. Space motion sickness, bone and muscle mass loss, cardiovascular deconditioning and neuro-ocular syndrome are some of the spaceflight-induced effects on human health. Optimal nutrition is of the utmost importance, and—in combination with other measures, such as physical activity and pharmacological treatment—has a key role in mitigating many of the above conditions, including bone and muscle mass loss. Since the beginning of human space exploration, space food has not fully covered astronauts’ needs. They often suffer from menu fatigue and present unintentional weight loss, which leads to further alterations. The purpose of this review was to explore the role of nutrition in relation to the pathophysiological effects of spaceflight on the human body.

## 1. Introduction

The space environment causes a plethora of stressors on the human body and has the potential to affect astronauts’ physiological and metabolic functions. Space stressors include space radiation, microgravity, isolation, and confinement [1,2,3]. These factors pose a challenge to the human body, which needs to adapt in order to cope with this hostile environment [4]. The future goals of human space exploration include missions to the Moon and Mars, but also space tourism [5,6]. Therefore, investigating the impact of spaceflight on the human body is of great significance, not only for astronauts, but also for the people who will travel to space as tourists. Furthermore, it will serve as a point of reference for future, long-term space missions—although current evidence is based on short-term missions so it should be carefully interpreted [1,7].

Optimal nutrition is of paramount importance for the crew’s health and the success of the mission, and an increasing number of studies have focused on the role of nutrition as a countermeasure to spaceflight-induced pathophysiological events [8,9]. Therefore, the aim of this review was to identify the effects of space stressors on astronauts’ physiology and to investigate the role of nutrition on human health under spaceflight conditions.

## 2. The Hostile Space Environment: Space Radiation, Microgravity, and Isolation

Space radiation is a primary risk factor for astronauts [10]. Exposure to ionizing radiation increases the production of reactive oxygen species (ROS) and leads to an imbalance between the production of oxygen free radicals and the body’s defensive antioxidant capacity, thus resulting in oxidative stress [8,11]. Microgravity is also a main risk factor for space travelers, as it provokes body fluid redistribution ‘’upwards’’ from the legs and abdomen to the heart and head. This causes multiple systems and organ deconditioning, including the vestibular, cardiovascular, and musculoskeletal system, as well as ophthalmic and endocrine changes, among others [4,12,13].

Other, less studied, yet principal factors include isolation and confinement. Living in an isolated and confined environment can cause social tensions, sleep disturbances, anxiety, and depression [14]. Furthermore, given space crews’ multinationalism, cultural differences among crew members on the International Space Station could also be a cause of psychosocial stress [15]. Moreover, the crew are not able to leave the spacecraft if they feel they need to abandon the mission, although this would be a rare event in the highly trained crews. These factors can lead to the appearance of pathophysiological signs and symptoms, from neurocognitive changes, and abnormal stress hormone levels to fatigue, sleep disturbances, and immunomodulatory changes [16,17]. As astronauts face abnormal day lengths and schedules, sleep disturbances may also result from a dysregulation of the 24 h cycle and consecutive changes in astronauts’ circadian rhythms [18]. Furthermore, anorexia and decreased energy intake have been observed during space missions, partially due to psychological factors [8]. Moreover, isolation, combined with exposure to microgravity and space radiation can induce significant alterations in the human body’s immune function [16].

The aforementioned effects vary greatly among individuals. Emotional instability, genetic predisposition, and personality differences can increase the susceptibility of the subjects to psychosocial, physiological, and sentimental changes [19,20]. Additionally, an astronaut’s experience has a significant impact on the solidarity of their behavior and on their overall adaptability to the new environment. For example, an experienced astronaut is less likely to face problems, compared to an astronaut on their first space mission [8,17].

Finally, contact with the natural environment on Earth, which has an undoubted effect on people’s mentality and quality of life, is not feasible in space. Neilson et al. [21] suggested ways to integrate a ‘’virtual’’ natural landscape into the International Space Station, using digital media, vegetation, virtual reality, and artificial windows in order to provide a virtual substitute to satisfy the astronauts’ instinctive need to be in contact with their natural environment.

The effect of spaceflight stressors on human health has gained scientific interest and multiple review papers have attempted to illuminate the metabolic, physiological, and mental disturbances experienced by astronauts due to the effects of the space environment, including radiation exposure, and sociological alterations resulting from their confinement and isolation [22,23]. In this narrative review, the physiological changes and the role of nutrition and nutritional status have been examined.

## 3. Spaceflight-Induced Physiological Changes

During the first days of spaceflight, space adaptation syndrome or space motion sickness is observed in astronauts. The main symptoms of the syndrome include nausea, dizziness, fatigue, anorexia, and headaches and they are attributed to the pathophysiological impairments in the vestibular system [8,24].

Moreover, the effects of the space environment on bone health has also been of scientific concern. During exposure to microgravity, bone resorption increases significantly, while bone formation remains unchanged or decreases. This imbalance leads to 1–1.5% bone mass loss per month [25]. This loss rate is similar to the bone loss rate observed in postmenopausal individuals per year [12]. Spaceflight-induced bone loss varies between skeletal sites. For example, bone tissue is better preserved in the non-weight-bearing upper limbs, compared to the weight-bearing lower limbs [26]. The increased level of bone resorption is associated with increased levels of calcium and other minerals in the blood circulation and the increased excretion of it in the urine, a process that enhances the astronauts’ risk of developing kidney stones during and after the spaceflight [27,28]. According to Gabel et al. [26], in a sample of seventeen astronauts (14 men and 3 women), spaceflight-induced bone loss was related to the duration of the flight; preflight bone turnover markers, which can be used to identify astronauts at risk; and pre-flight physical activity levels. Further research is needed to investigate whether this loss is recuperated upon their return to Earth. Gabel et al. [29] observed an incomplete bone loss reversal in astronauts’ tibia 1 year from the end of their spaceflight. It is believed that flight duration is proportional to the reverse of bone loss. The longer the space flight, the greater the loss and the more difficult the recovery. 

Apart from bone loss, muscle loss and a rapid atrophy of skeletal muscle is also observed in astronauts, as, under microgravity conditions, movements require minimal muscle effort and force, which causes muscle mass and volume loss, especially in the lower limbs [30,31]. This decrease in muscle mass leads to weakness and diminished functional capacity, which is particularly noticeable in astronauts upon their return to a gravity condition [32]. A recent review of the factors directly contributing to this phenomenon include a change in the levels of cortisol, vitamin D, antioxidants, growth hormones, thyroid hormones, and testosterone, which occur during space flights [33]. Studies have shown that rapid volumetric skeletal muscle loss, as revealed by magnetic resonance imaging (MRI), occurs in the quadriceps (−6%); gastrocnemius (−6%); and posterior back muscles (−10%), following 6–9 days of spaceflight [34,35].

The cardiovascular system is also influenced by spaceflight. Body fluid redistribution due to microgravity leads to approximately 2 L of fluid shifting to the upper limbs and head, thus increasing cardiac output (CO) by 18–26% [15,36]. The aforementioned fluid shift causes a decrease in the circulating blood volume (10–15%), aerobic capacity, and heart size (cardiac atrophy) [37,38]. These changes are associated with orthostatic hypotension, a phenomenon that most astronauts experience after a spaceflight [15,39]. Moreover, according to a prospective cohort study of 11 astronauts during long-duration missions on the International Space Station, microgravity was associated with a cessation of blood flow in the internal jugular vein, which can, in turn, lead to thrombosis in otherwise healthy astronauts [40]. Thrombosis is a phenomenon that needs further study, as there is evidence of microgravity-induced changes in hemostasis mechanisms, such as decreased blood velocity, endothelial dysfunction, and increased fibrinogen synthesis [41,42]. 

Another major consequence of space travel is the dysregulation of the immune system. It is well-established that the immune system is influenced by various factors, including environment, diet, microbiome, and psychological stressors. During short-term six-month flights, astronauts are exposed to space radiation, face circadian rhythm dysregulations, and micronutrient deficiencies, among others, which are all to blame for astronauts’ altered immunity [40]. Prolonged isolation and confinement may also lead to reduced immunity [16]. Adequate nutritional intake is crucial for an optimal immune function, but no specific nutrient has proved to be effective for this purpose. 

The ocular system and vision are also negatively affected during space travel. Previously published data suggest that astronauts experience vision changes; distance and near visual acuity deterioration; and the so-called spaceflight associated neuro-ocular syndrome (SANS), for which is mainly characterised by optic disc edema [42].

Finally, other factors include insulin resistance, impaired wound healing, and alterations in the respiratory and the central nervous system [42,43,44]. 

## 4. Nutritional Concerns during Spaceflights

The main nutritional concerns during a spaceflight include the sufficient provision of energy to counteract the negative energy balance, which is often experienced by astronauts, the prevention of a deficiency in micronutrients, and fluid and sodium management [5,9]. Weight loss, which has been observed since the early space exploration missions, is a major health concern for astronauts and a main factor leading to accelerated muscle mass loss [45].

Low energy intake is negatively associated with cardiovascular deconditioning, according to bed rest studies that have investigated the effect of hypocaloric diets on astronauts’ physiology [46,47]. During future long-term missions, negative energy balance will possibly be a cause of greater implications [48]. Factors that may cause reduced dietary intake include altered taste and smell due to the redistribution of body fluids, limited food variety and palatability, isolation, menu fatigue, altered microbiome, and increased physical activity levels [48,49,50]. These factors are depicted in Figure 1.

Furthermore, according to a recent systematic review by Tang et al. [5], the high concentration of carbon dioxide on the space stations affects energy intake by reducing food consumption. However, the easy preparation of food, i.e., the simple addition of hot water, is positively linked to the food consumption of astronauts [45]. Strategies to mitigate this problem are the addition of fresh, tasty, calorie-dense foods to the menu, considering food culture and cultural habits and the promotion of joint eating activities among astronauts. The environmental and physiological factors that affect the smell and taste of astronauts during a space flight need further study [5,45,49].

Regarding the deficiency of micronutrients, copper and zinc require extra attention, as a 21-day bed rest study showed increased fecal excretion of these micronutrients [51]. Regarding sodium levels, most space foods are high in salt due to the processing and the need for long-term preservation. This high sodium intake can have several negative pathophysiological effects, such as elevated urinary calcium excretion and an increased risk of kidney stones. Despite efforts to limit added in food items consumed by the participants in a spaceflight, current food on the ISS stillcontains high amounts of salt. A recent review by Tang et al. [5] supported the replacement of high sodium processed food items with fresh ones, which should be preserved in appropriate conditions to ensure their palatability and the preservation of their organoleptic properties. 

## 5. Space Food Evolution and Quality Assessment

Space Food is a term used to describe the food items that astronauts consume on the International Space Station. Space food during early space missions included limited options and it was consumed through small tubes, while, nowadays, astronauts enjoy a variety of options, which have been specially processed to meet specific requirements [52,53]. Currently, the foods on the ISS are in a rehydratable, thermostabilized, and natural form. There are also irradiated and fresh options [53,54].

As astronauts’ access to fresh food relies on cargo resupplies from Earth, this is relatively limited, a condition which has been linked to a higher constipation risk among astronauts [14]. In more distant future missions, such as missions to Mars, resupplies with fresh food from Earth will not be an option. An alternative scenario could be the inclusion of functional foods, which contain nutrients that can potentially offer an enhanced nutrient content and related health benefits to astronauts. Possible nutrients that could be used in functional foods are omega-3 fatty acids, anthocyanins, and glutathione [5,40].

Space food is stored for months on the ISS and exposed to space radiation, which can potentially alter its quality and stability [54]. Zwart et al. [55] sent five food items that were combined with a multivitamin and a vitamin D supplement to the International Space Station for 13, 353, 596, and 880 days and analyzed them after their return to Earth. The researchers observed vitamin degradation after one or more years of storage, but the total change was not significantly different from the controls on Earth, leading to the conclusion that storage duration, and not spaceflight per se, is the principal factor leading to degradation [55].

A more recent study [54], which included a wider variety of foods (109 available items), concluded that vitamin D, vitamin K, potassium, and calcium concentrations were insufficient to meet the recommended daily intake requirements, even before storage. Regarding vitamin degradation due to prolonged storage, the most unstable vitamins were B1 and C, which presented lower degradation efficiency after 1 and 3 years, respectively. Therefore, further studies are needed regarding the nutritional content of the menu, as well as the nutrient stability for long-term space missions.

Extensive efforts have been made to create a sustainable food production system, which will be sufficient and efficient for long-duration manned space missions [45]. A diet based on plant-based foods is promising and includes plant cultivation in the spacecraft, something that has already been applied, and that has proved to be effective on the ISS [56]. According to a recent review [45], space food systems should be characterized by safety, reliability, stability, palatability, variety, and taste.

## 6. The Role of Nutrition in Space-Induced Pathophysiological Effects

Astronauts’ diets consist of 55% carbohydrates (CHO), 30% fat, and 15% protein (PRO), and are based on a 4-to-6-day cycle menu [5,57]. A high-protein nutritional regimen (45:25:30 CHO:FAT:PRO) led to greater performance in a sample of 22 astronauts and could be considered for longer space missions [58]. Nutrition could act as a countermeasure against spaceflight-induced pathophysiological adaptations [36,59,60].

### 6.1. The Role of Antioxidants against the Effects of Space Radiation 

Prolonged exposure to space radiation can lead to alterations in the central nervous system and in cognitive and cardiovascular function and may induce DNA damage and enhance free radical production [61]. According to a recent editorial [62], this phenomenon is attributed to increased high-energy radiation, microgravity, hypoxia, and hyperoxia during extravehicular activities (EVAs) and can result from the simultaneous overexposure to multiple toxic factors. In long-term space missions, the inhalation of lunar and Martian dust and regolith is also possible [62]. These two factors (radiation and microgravity) continue to be routinely studied, and there are considerable data in the general scientific literature reflecting their impact on oxidative stress and damage (OSaD) coefficients through genomic, proteomic, and metabolomic responses [63]. However, current literature has not investigated variances in gas content and pressures, and the induction of probably poisonous dust/regolith debris into the body.

Antioxidant-rich foods could be an alternative to mitigate the space radiation-induced effects [5,23,64]. There is limited evidence that supports the fact that sulfur-containing antioxidants reduce the levels of oxygen free radicals, and research is needed to verify the possible role of antioxidants against space radiation-induced consequences [65].

### 6.2. Nutritional Countermeasures against Bone and Muscle Mass Loss

Another major consequence of space travel is bone loss, which has been documented since the first space exploration missions (Figure 2) [66,67]. Calcium and vitamin D consumption are of primary importance, as low levels of these nutrients may accelerate bone loss. However, high doses of calcium and vitamin D by dietary supplements have not been proven to be beneficial against bone loss [59], as calcium excretion—which is a result of bone loss in microgravity environments—may lead to the suppression of parathyroid hormone, which, in turn, decreases the absorption of calcium and vitamin D levels [68]. Vitamin D deficiency among astronauts may be attributed to the lack of natural sunlight; therefore, astronauts have to rely on dietary sources and supplements of vitamin D [69]. Astronauts require an adequate oral intake of vitamin D and Ca^2+^ during spaceflights, and the amount necessary to minimize a negative balance is approximately 1000 mg/d of calcium and 800–1000 IU/d of vitamin D, which is currently recommended for space flights of up to one year [69].

There is also evidence that one potential nutritional countermeasure against bone loss is lowering sulfur-containing amino acids and sodium chloride consumption, which seem to induce a low-level metabolic acidosis, a phenomenon that might cause increased bone resorption. Given that space food is very high in salt, sodium chloride is a potential factor that contributes to astronauts’ bone loss and its reduction is a potentially effective preventive measure [68].

In regard to fatty acids, omega-3 supplementation has also been linked to positive outcomes on bone metabolism [70]. Protein is also essential for bone and muscle health. However, the use of protein and amino acid supplements have long been studied as a possibility to mitigate muscle loss, however, the results have shown no effectiveness in the absence of physical exercise [71]. Glucose intolerance is a complication observed both during spaceflight and in bed rest studies. The application of low glycemic index diets could mitigate spaceflight-induced insulin resistance, and also cardiovascular alterations [59].

Dorfman et al. [72] used an approach of combining physical activity and branched-chain amino acid supplementation in a randomized clinical bed rest trial and this combination was proven to be effective for the mitigation of cardiac atrophy. Physical exercise is a part of the astronauts’ routine and includes both aerobic and anaerobic training. Currently, astronauts use the advanced resistance exercise device (ARED), which has been developed to counter spaceflight-induced muscle atrophy. However, recent data have cast doubt on the efficiency of this device. More specifically, a systematic review by Comfort et al. concluded that the ARED device is not an effective countermeasure for mitigating the effects of microgravity on muscles [30]. Therefore, a review of astronauts’ current exercise program and the development of more efficient protocols is required, especially for longer space exploration missions.

### 6.3. Gut Microbiome and the Role of the Supplementation of Probiotics 

The space environment imposes several challenges to the human microbiota [73]. Recent studies have revealed that changes in gut microbiota during spaceflight could potentially influence not only nutrient assimilation and energy intake, but also the function of the immune system [9,74,75].

Mounting evidence has highlighted spaceflight stressor-induced microbiome dysbiosis and changes in the composition of the intestinal, nose, tongue, and skin microbiota [73,76,77,78,79]. The intake of probiotics in the form of dietary supplements could be potentially beneficial for astronauts’ microbiome, and a promising countermeasure against space-induced immune changes [9,80]. Previous findings suggest that the Lactobacillus casei strain Shirota (LcS) may improve innate immunity [81,82]. Based on this, Sakai et al. [83] developed a probiotic capsule, containing freeze-dried LcS and sent it to the ISS, in order to investigate its stability during 1 month of storage under spaceflight conditions. The product’s basic probiotic properties were maintained and the authors stated that it may be safely used during space missions. Other initiatives included the creation of a calcium-rich, freeze-dried yogurt. Venir et al. [84] observed that the freeze-drying process caused a reduction in the lactic acid bacteria population, something which was countered by the addition of blueberries and sucrose in the product. Finally, in a recent review paper, Turroni et al. [80] concluded that, although the administration of probiotics is non-interventional and potentially beneficial, further studies are needed to prove their efficacy.

A short summary of the proposed nutritional countermeasures against spaceflight-induced effects is presented in Table 1.

## 7. Benefits of Space Research on Earth

Space research benefits both space travelers and people on Earth [9,27]. This has been already proven, as there are examples of technologies that were first used in space, but which then proved to be useful on Earth too, such as echocardiography and the hazard analysis and critical control points’ (HACCP) application [85,86].

Furthermore, previous research has indicated that the pathophysiological adaptations observed during space missions are equivalent to an accelerated aging process [87,88]. Therefore, space exploration can be particularly beneficial for public health. Investigating efficient countermeasures to prevent and treat muscle atrophy, a typical aging-like effect of microgravity, could be beneficial on Earth for the management of sarcopenia in the elderly, while measures against spaceflight-induced bone loss could be applied to patients with osteopenia and osteoporosis [9].

Finally, current research on the development of a sustainable food production system for longer space missions might benefit sustainability goals on Earth [45].

## 8. Conclusions

Spaceflight stressors have been associated with pathological changes in astronauts’ physiology and metabolism. The role of nutrition—although possibly not as a standalone countermeasure—appears to be crucial for a crew’s health as it can potentially mitigate space-induced health effects, such as bone loss and microbiota dysbiosis.

Data from early space missions showed astronauts’ inability to meet the required daily energy needs, possibly due to altered tastes, low food palatability and variety, excess physical activity, or isolation. This negative energy balance interferes with other pathophysiological adaptations, including mass loss and cardiovascular deconditioning. In this context, space menus need further modifications in terms of nutrient content, stability, and taste.

Finally, given that most evidence is based on short-term space missions, current ambitions for future explorations of the Moon and Mars require multidisciplinary efforts and studies to find the most optimal medical and nutritional protocols to ensure crews’ health and well-being. Thus, the role of a specialized dietitian within the space medical team is of primary importance.

## Figures and Tables

**Figure 1 nutrients-14-04896-f001:**
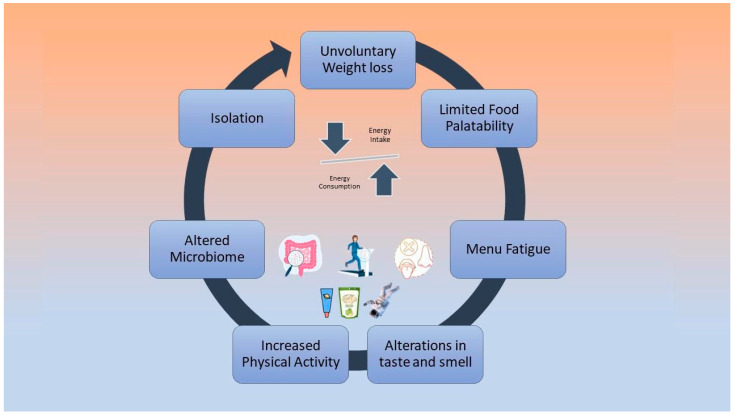
Causes of negative energy balance during spaceflights.

**Figure 2 nutrients-14-04896-f002:**
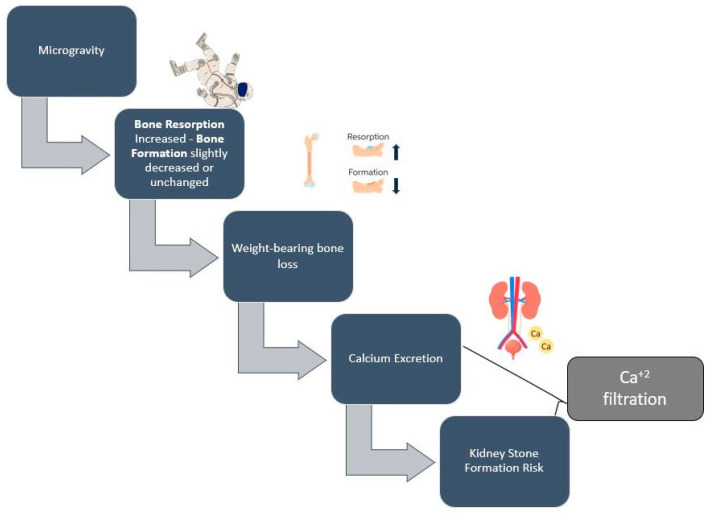
Simplified proposed mechanism of spaceflight-induced bone loss and kidney stone formation risk.

**Table 1 nutrients-14-04896-t001:** Proposed Nutritional Countermeasures Against Spaceflight-Induced Effects on Astronauts’ Health.

Spaceflight Effect	Proposed Countermeasures	Proposed Nutritional Countermeasures
Oxidative Stress	Pharmaceutical Nutritional Materials against radiation for spaceships and spacesuits	Natural antioxidants, sulfur-containing amino acids, Omega-3 fatty acids, vitamin E, vitamin C, beta-carotene, and selenium [4,5,40,64].
Bone Loss	Pharmaceutical Nutritional Physical exercise	Protein, calcium, Vitamin D, Vitamin K, omega-3 fatty acids, and vitamin E [59].
Muscle Atrophy	Nutritional Physical exercise	BCAA supplementation [59].
Microbiota Alterations	Nutritional Pharmaceutical	Probiotics [80,84].
Negative Energy Balance	Nutritional	Energy-dense, palatable food; bigger variety; joint meal activities among the crew; and consideration of food culture [49,50].

## Data Availability

Not applicable.

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
