# Peer review of "Physiological Alterations in Relation to Space Flight: The Role of Nutrition"

_nutrients, 2022, doi:10.3390/nu14224896_

Round 1

Reviewer 1 Report

General comments

This manuscript aims at identifying the effects of space stressors on astronauts’ physiology and investigating the role of nutrition on human health under spaceflight conditions. Authors manage to fulfill sufficiently their aims. Yet, to get acceptance recommendation, they should address several format issues detailed below.

Minor comments

(line 25) … confinement [1-3]. (viz., missing space, as well. This elsewhere throughout MS, too)

(l62) … experience has a… (i.e., there is a bold “s”)

(l65) … an undoubted effect on human’s…

(l66) … mentality, and… (i.e., there is a red comma)

(l95) Gabel et al…

(l125) “edemaclude”?

Please, introduce Figure 1 before showing it;

(l143 and elsewhere throughout MS) … by Tang et al…

(l175) … al. [54] sent…

(l179) … storage, but total change…

(l183) … and calcium concentrations were…

(l186) … which presented lower degradation efficiency…

(l208) … possible [11,62]. (i.e., Word Track Changes remain)

(l208) please, introduce μG (not all readers may know this acronym);

(l211) … in gas content…

(l212) … of probably poisonous dust…

(l213) … with inside the literature.

I STOP HIGHLIGHTING THESE MISSING SPACE (FURTHER ARE THERE) (Windows Word issue?)

Please, introduce Table I before showing it;

(l278) Table I: Proposed… (i.e., “T” not bold)

(Table I) Word Track Changes remain.

Author Response

Reply to reviewers

Reviewer 1

General comments

This manuscript aims at identifying the effects of space stressors on astronauts’ physiology and investigating the role of nutrition on human health under spaceflight conditions. Authors manage to fulfill sufficiently their aims. Yet, to get acceptance recommendation, they should address several format issues detailed below.

Answer: We would like to thank reviewer #1 for the kind remark. We will try to address all the comments one by one.

Minor comments

(line 25) … confinement [1-3]. (viz., missing space, as well. This elsewhere throughout MS, too)

Answer: All the relative changes have been made.

(l62) … experience has a… (i.e., there is a bold “s”)

Answer: it has been corrected.

(l65) … an undoubted effect on human’s…

Answer: it has been corrected.

(l66) … mentality, and… (i.e., there is a red comma)

Answer: it has been corrected.

(l95) Gabel et al…

Answer: it has been corrected.

(l125) “edemaclude”?

Answer: it has been corrected and now line 106 is the following; being the optic disc edema, clude dysregulation of the immune system,…

Please, introduce Figure 1 before showing it;

Answer: Figure 1 is mentioned in Line 122, just before p 4 where Figure 1 is shown

(l143 and elsewhere throughout MS) … by Tang et al…

Answer: All the relative changes have been made.

(l175) … al. [54] sent…

Answer: it has been corrected.

(l179) … storage, but total change…

Answer: it has been corrected.

(l183) … and calcium concentrations were…

Answer: it has been corrected.

(l186) … which presented lower degradation efficiency…

Answer: it has been corrected.

(l208) … possible [11,62]. (i.e., Word Track Changes remain)

Answer: it has been corrected.

(l208) please, introduce μG (not all readers may know this acronym);

Answer: Relevant information has been added (line 189).

(l211) … in gas content…

Answer: it has been corrected.

(l212) … of probably poisonous dust…

Answer: it has been corrected.

(l213) … with inside the literature.

Answer: it has been corrected.

I STOP HIGHLIGHTING THESE MISSING SPACE (FURTHER ARE THERE) (Windows Word issue?)

Answer: We are sorry for that problem. It has been probably caused during the upload of the file. We tried to correct all the problematic areas.

Please, introduce Table I before showing it;

Answer: Thank you for the comment. A relative sentence introducing table I has been added. Line 232: A short summary of the proposed nutritional countermeasures against spaceflight-induced effects is presented in Table I.

I.

(l278) Table I: Proposed… (i.e., “T” not bold)

Answer: it has been corrected.

(Table I) Word Track Changes remain.

Answer: it has been corrected.

Reviewer 2 

This is a review article on the effect of space flight on human physiology and the role of nutrition. The manuscript is well organized, and the content of this review is comprehensive, including 87 references in extensive research areas. Here are a few items that can be included to improve this review.

Major

  • Space flight: the conditions may differ depending on the type of space flight, including low-altitude earth-rotating orbits like a space station, moon landing, Mars travel, and beyond. The radiation levels from the Sun, the 3D space recognition, light/dark cycles, etc. It is recommended to mention the scope of this review.

 Answer: Thank you for the comment. A relative sentence been added in Line 255

  • Immune responses: This review does not describe much about the immune responses, but the alterations in immunity in space are one of the major physiological alterations. It is recommended to describe the effects on immunity and the role of nutrition.

 Answer: Thank you for the comment. A relative paragraph been added in Lines 98-103

  • 24 h cycle: This cycle can be disrupted, and its effects can be described.

Answer: Thank you for the comment. A relative paragraph been added in Lines 45-47

Minor

  • Please proofread the manuscript. For instance, lines 65, 66.

Answer: We are sorry for that problem. It has been probably caused during the upload of the file. We tried to correct all the problematic areas.

  • Figures 1 and 2: the small illustrations such as an astronaut, bone, etc. can be enlarged. Please make sure that the illustrations are original.

Answer: All illustrations are original

Reviewer 2 Report

This is a review article on the effect of space flight on human physiology and the role of nutrition. The manuscript is well organized, and the content of this review is comprehensive, including 87 references in extensive research areas. Here are a few items that can be included to improve this review.

Major

·         Space flight: the conditions may differ depending on the type of space flight, including low-altitude earth-rotating orbits like a space station, moon landing, Mars travel, and beyond. The radiation levels from the Sun, the 3D space recognition, light/dark cycles, etc. It is recommended to mention the scope of this review.

·         Immune responses: This review does not describe much about the immune responses, but the alterations in immunity in space are one of the major physiological alterations. It is recommended to describe the effects on immunity and the role of nutrition.

·         24 h cycle: This cycle can be disrupted and its effects can be described.

Minor

·         Please proofread the manuscript. For instance, lines 65, 66.

·         Figures 1 and 2: the small illustrations such as an astronaut, bone, etc. can be enlarged. Please make sure that the illustrations are original.

Author Response

(The authors gave the same response as above.)
